Detecting communicative intent in a computerised test of joint attention

Caruana Nathan nathan.caruana@mq.edu.au 1 2 3
McArthur Genevieve 1 2 4
Woolgar Alexandra 1 2 3
Brock Jon 2 4 5
1 Department of Cognitive Science, Macquarie University , Sydney , NSW , Australia
2 ARC Centre of Excellence in Cognition and its Disorders , Sydney , NSW , Australia
3 Perception in Action Research Centre , Sydney , NSW , Australia
4 Centre for Atypical Neurodevelopment , Sydney , NSW , Australia
5 Department of Psychology, Macquarie University , Sydney , NSW , Australia
Borghi Anna
Electronic publication date: 2017 Jan 17
Publication date: 2017
Volume: 5
Electronic Location ID: e2899
Received 2016 Aug 31; Accepted 2016 Dec 12
Copyright: ©2017 Caruana et al.
Copyright year: 2017
Copyright holder: Caruana et al.
License: This is an open access article distributed under the terms of the Creative Commons Attribution License, which permits unrestricted use, distribution, reproduction and adaptation in any medium and for any purpose provided that it is properly attributed. For attribution, the original author(s), title, publication source (PeerJ) and either DOI or URL of the article must be cited.
License URL: https://creativecommons.org/licenses/by/4.0/

Keywords: Joint attention, Social interaction, Eye-tracking, Virtual reality, Eye gaze, Mentalising

Funding: Australian Research Council CE110001021 DE120100898 DP098466 Australian Postgraduate Award This work was supported by the Australian Research Council [CE110001021, DE120100898, DP098466] and an Australian Postgraduate Award to Dr. Caruana. The funders had no role in study design, data collection and analysis, decision to publish, or preparation of the manuscript.

==============================
The successful navigation of social interactions depends on a range of cognitive faculties—including the ability to achieve joint attention with others to share information and experiences. We investigated the influence that intention monitoring processes have on gaze-following response times during joint attention. We employed a virtual reality task in which 16 healthy adults engaged in a collaborative game with a virtual partner to locate a target in a visual array. In the Search task, the virtual partner was programmed to engage in non-communicative gaze shifts in search of the target, establish eye contact, and then display a communicative gaze shift to guide the participant to the target. In the NoSearch task, the virtual partner simply established eye contact and then made a single communicative gaze shift towards the target (i.e., there were no non-communicative gaze shifts in search of the target). Thus, only the Search task required participants to monitor their partner’s communicative intent before responding to joint attention bids. We found that gaze following was significantly slower in the Search task than the NoSearch task. However, the same effect on response times was not observed when participants completed non-social control versions of the Search and NoSearch tasks, in which the avatar’s gaze was replaced by arrow cues. These data demonstrate that the intention monitoring processes involved in differentiating communicative and non-communicative gaze shifts during the Search task had a measurable influence on subsequent joint attention behaviour. The empirical and methodological implications of these findings for the fields of autism and social neuroscience will be discussed.

Introduction

Joint attention is defined as the simultaneous coordination of attention between a social partner and an object or event of interest (Bruner, 1974; Bruner, 1995). It is an intentional, communicative act. In the prototypical joint attention episode, one person initiates joint attention (IJA) by pointing, turning their head, or shifting their eye gaze to intentionally guide their social partner to an object or event in the environment. The partner must recognise the intentional nature of this initiating behaviour and respond to that joint attention bid (RJA) by directing their attention to the cued location (Bruinsma, Koegel & Koegel, 2004).

The ability to engage in joint attention is considered critical for the normal development of language and for navigating social interactions (Adamson et al., 2009; Charman, 2003; Dawson et al., 2004; Mundy, Sigman & Kasari, 1990; Murray et al., 2008; Tomasello, 1995) and its developmental delay is a hallmark of autism spectrum disorders (Lord et al., 2000; Stone, Ousley & Littleford, 1997). Yet despite its importance to both typical and atypical development, very little is known about the neurocognitive mechanisms of joint attention. By definition, joint attention involves an interaction between two individuals. The challenge for researchers, therefore, has been to develop paradigms that achieve the ecological validity of a dynamic, interactive, social experience, whilst at the same time maintaining experimental control.

In a recent functional magnetic resonance imaging (fMRI) study, Schilbach et al. (2010) investigated the neural correlates of joint attention using a novel virtual reality paradigm. During the scan, participants’ eye-movements were recorded as they interacted with an anthropomorphic avatar. They were told that the avatar’s gaze was controlled by a confederate outside the scanner also using an eye-tracking device. In fact, the avatar was controlled by a computer algorithm that responded to the participant’s own eye-movements. On RJA trials (referred to as OTHER_JA by Schilbach et al., 2010), the avatar looked towards one of three squares positioned around his face, and participants were instructed to respond by looking at the same square. Participants also completed IJA trials in which the roles were reversed.

Similar tasks have been used in other fMRI studies using either gaze-contingent avatars (Oberwelland et al., in press) or live-video links to a real social partner (Redcay et al., 2012; Saito et al., 2010). Together, these interactive paradigms represent an important step towards an ecologically valid measure of joint attention. There is, however, a potentially important limitation of the tasks used in these studies: in each task, every trial involved a single unambiguously communicative eye-gaze cue. On RJA trials, the participant’s partner would make a single eye-movement towards the target location and the participant knew they were required to respond to that isolated cue. This differs from real-life joint attention episodes, which are embedded within complex ongoing social interactions. In real life, responding to a joint attention bid requires that the individual first identifies the intentional nature of their partner’s behaviour. That is, they must decide whether or not the cue is one they should follow. We refer to this component of joint attention as “intention monitoring.”

In a recent fMRI study, we developed a novel joint attention task to better capture this intention monitoring process (Caruana, Brock & Woolgar, 2015). Following Schilbach et al. (2010), participants played a cooperative game with an avatar whom they believed to be controlled by a real person (referred to as “Alan”), but was actually controlled by a gaze-contingent algorithm. The participant and avatar were both allotted onscreen houses to search for a burglar (see Fig. 1). On IJA trials, the participant found the burglar, made eye contact with the avatar, and then guided the avatar to the burglar by looking back at the house in which the burglar was hiding. On RJA trials, the participant found all of their allotted houses to be empty. Once Alan had finished searching his own houses, he would make eye contact with the participant before guiding them towards the house containing the burglar.

Figure 1 Experimental display showing the central avatar (“Alan”) and the six houses in which the burglar could be hiding.

Gaze areas of interest (GAOIs), are represented by blue rectangles, and were not visible to participants.

The critical innovation of this task was the initial search phase. This provided a natural and intuitive context in which participants could determine, on each trial, their role as either the responder or initiator in a joint attention episode (previous studies had provided explicit instructions; e.g., Schilbach et al., 2010; Redcay et al., 2012). More importantly for current purposes, the RJA trials required participants to monitor their partner’s communicative intentions. During each trial, the avatar made multiple non-communicative eye-movements as he searched his own houses. The participant had to ignore these eye-movements and respond only to the communicative “initiating saccades” that followed the establishment of eye contact. This is consistent with genuine social interactions in which eye contact is used to signal one’s readiness and intention to communicate, particularly in the absence of verbal cues (Cary, 1978).

We compared the RJA trials in this new paradigm to non-social control trials (referred to as RJAc) in which the eye gaze cue was replaced by a green arrow superimposed over the avatar’s face. Analysis of saccadic reactions times revealed that participants were significantly slower (by approximately 209 ms) to respond to the avatar’s eye gaze cue than they were to respond to the arrow (Caruana, Brock & Woolgar, 2015). This effect was surprising—previous studies have shown that gaze cues often engender rapid and reflexive attention shifts (see Frischen, Bayliss & Tipper, 2007), but that would predict faster rather than slower responses to gaze cues. Nevertheless, we have since replicated this finding in an independent sample of adults and, intriguingly, found that the effect is exaggerated in a group of autistic individuals (Caruana et al., in press).

One explanation for these findings is that they reflect the intention monitoring aspects of RJA. Specifically, participants are slower to respond to eye gaze cues than arrows because it takes time to identify the cue as being an intentional and communicative bid to initiate joint attention. In the control condition, the arrow presents an unambiguous attention cue, and so the participant does not need to decide whether they should respond to it or not. The implication here is that intention monitoring is a cognitively demanding operation that requires time to complete and is manifest in the response times to eye gaze cues.

However, before reaching such a conclusion, it is important to consider a number of alternative explanations. For example, it may be that participants responded faster in the RJAc condition because the large green arrow cue, which extended towards the target location, provided a more salient spatial cue than the avatar’s eyes. It is also possible that the mere context of social interaction may influence the way participants approach the task. In particular, when individuals believe they are interacting with an intentional human agent, mirroring and mentalising mechanisms are automatically recruited which exert a top-down effect on the neural processes governing visual perception or attention (Wykowska et al., 2014; Caruana, De Lissa & McArthur, 2016).

The aim of the current study, therefore, was to test the intention monitoring account more directly by manipulating the intention monitoring component of the RJA task whilst controlling for both the perceptual properties of the stimulus and the social nature of the task. To this end, we tested a new sample of participants using the same task but with one further manipulation. On half the trials, we eliminated the search phase of the task. Thus, on RJA trials, the avatar only made a single eye movement to the target to initiate joint attention, and participants knew unambiguously that they should follow it. The gaze cues in the ‘Search’ and ‘NoSearch’ versions of the task were identical and in both cases, participants believed they were interacting with another human. Thus, only the intention monitoring account predicts an effect of task (Search versus NoSearch) on response times. Participants also completed Search and NoSearch versions of the control (RJAc) condition. Because the arrow cue is unambiguous whether or not it is preceded by a search phase, we did not predict any difference in response times. In other words, a condition (Social vs. Control) by task (Search vs. NoSearch) interaction would indicate that response times to joint attention bids are influenced by the intention monitoring processes that precede true RJA behaviours.

Method

Ethical statement

The study was approved by the Human Research Ethics Committee at Macquarie University (MQ; reference number: 5201200021). Participants received course credit for their time and provided written consent before participating.

Participants

Sixteen right-handed adults with typical development, normal vision, and no history of neurological impairment participated in this study (three female, Mage = 19.92, SD = 1.03).

Stimuli

We employed an interactive paradigm that we had previously used to investigate the neural correlates of RJA and IJA (Caruana, Brock & Woolgar, 2015). The stimuli comprised an anthropomorphic avatar face, generated using FaceGen (Singular Inversions, 2008), that subtended 6.5 degrees of visual angle. The avatar’s gaze was manipulated so that it could be directed either at the participant or towards one of the six houses that were presented on the screen (see Fig. 1). The houses were arranged in two horizontal rows above and below the avatar and each subtended four degrees of visual angle.

Joint attention task

Social Conditions (RJA and IJA)

Participants played a cooperative “Catch-the-Burglar” game with an avatar whom they believed was controlled by another person named “Alan” in a nearby eye tracking laboratory using live infrared eye tracking. In reality, a gaze-contingent algorithm controlled the avatar’s responsive behaviour (see Caruana, Brock & Woolgar, 2015) for a detailed description of this algorithm and a video demonstration of the task). The goal of the game was to catch a burglar that was hiding inside one of the six houses presented on the screen. Participants completed two versions of the social conditions (i.e., Search and NoSearch tasks) during separate blocks.

Search task

This task was identical to the “Catch-the-Burglar” task employed in our previous work (e.g., Caruana, Brock & Woolgar, 2015). Each trial in the Search task began with a “search phase”. During this period, participants saw two rows of houses on a computer screen including a row of three blue doors and a row of three red doors. They were instructed to search the row of houses with blue doors while Alan searched the row of houses with red doors. Participants were told that they could not see the contents of Alan’s houses and that Alan could not see the contents of their houses. Whoever found the burglar first had to guide the other person to the correct location.

Participants searched the houses with blue doors in any order by fixating on them. Once a fixation was detected on a blue door, it opened to reveal either the burglar or an empty house. On some trials, only one or two blue doors were visible, whilst the remaining doors were already open. This introduced some variability in the order with which participants searched their houses that made Alan’s random search behaviour appear realistically unpredictable.

Once the participant fixated back on the avatar’s face, Alan was programmed to search 0–2 more houses and then make eye contact. This provided an interval in which participants could observe Alan’s non-communicative gaze behaviour as he completed his search. The onset latency of each eye movement made by Alan was jittered with a uniform distribution between 500–1,000 ms.

On RJA trials, participants discovered that all of their allotted houses were empty (Fig. 2, row 1), indicating that the burglar was hiding in one of Alan’s houses. Once the participant fixated back on Alan’s face, he searched 0–2 more houses in random order before establishing eye contact with the participant. Alan then initiated joint attention by directing his gaze towards one of his allotted houses. If the participant made a “responding saccade” and fixated the correct location, the burglar was captured.

Figure 2 Experimental task event timeline.

Event timeline for RJA and RJAc trials.

On IJA trials, the participant found a burglar behind one of the blue doors. They were then required to fixate back on Alan’s face, at which point the door would close again to conceal the burglar. Again, Alan was programmed to search 0–2 houses before looking straight at the participant. Once eye contact was established, participants could initiate joint attention by making an “initiating saccade” to fixate on the blue door that concealed the burglar. Alan was programmed to only respond to initiating saccades that followed the establishment of eye contact, and to follow the participant’s gaze, irrespective of whether the participant fixated the correct house or not. Whilst performance on IJA trials was not of interest in the current study, the inclusion of this condition created a context for the collaborative search element of the task and allowed direct comparison with our previous studies in which participants alternated between initiating and responding roles.

When the participant made a responding or initiating saccade to the correct location, the burglar appeared behind prison bars to indicate that he had been successfully captured (e.g., Fig. 2, column 7). However, the burglar appeared in red at his true location, to indicate that he had escaped, if participants (1) made a responding or initiating saccade to an incorrect location, (2) took longer than three seconds to make a responding or initiating saccade, or (3) spent more than three seconds looking away from task-relevant stimuli (i.e., Alan and houses). Furthermore, trials were terminated if the participants took longer than three seconds to begin searching their houses at the beginning of the trial. On these trials, red text reading “Failed Search” appeared on the screen to provide feedback.

NoSearch task

This version of the task was identical to the Search task except that the search phase in each trial was removed. In IJA trials, all but one house was visibly empty (i.e., the door was open and no burglar was present), and participants were instructed that if they saw a blue door in their allotted row of houses that the burglar would be “hiding” behind it. In RJA trials, all of the houses were visibly empty. For both IJA and RJA trials, Alan’s eyes would be closed at the beginning of the trial, and then open after 500–1,000 ms (jittered with a uniform distribution) so that he was looking at the participant. Alan would then wait to be guided on IJA trials. On RJA trials, Alan shifted his gaze to guide the participant after a further 500–1,000 ms, provided that eye contact had been maintained. Thus, in both the Search and NoSearch tasks, Alan made eye contact with the participant before guiding them to the burglar on RJA trials. Therefore, the perceptual properties of the gaze cue itself were identical between tasks, but the NoSearch task removed the requirement to use the eye contact cue to identify communicative gaze shifts.

Control Conditions (RJAc and IJAc)

For each of the social conditions in both versions of the task, we employed a control condition that was closely matched on non-social task demands (e.g., attentional orienting, oculomotor control). In these conditions (RJAc and IJAc), participants were told that they would play the game without Alan, whose eyes remained closed during the trial. Participants were told that the stimuli presented on the computer screen in these trials were controlled by a computer algorithm. In the Search task, a grey fixation point was presented over the avatar’s nose until the participant completed their search and fixated upon it. After a short delay, the fixation point turned green (analogous to the avatar making eye contact). From this point onwards, the Search and NoSearch tasks were identical. On IJAc trials, the green fixation point was the cue to saccade towards the burglar location. On RJAc trials a green arrow subtending three degrees of visual angle cued the burglar’s location (analogous to Alan’s guiding gaze; see Caruana, Brock & Woolgar, 2015 for a video with example trials from each condition).

Procedure

Joint attention task

The experiment was presented using Experiment Builder 1.10.165 (SR Research, 2004). Participants completed four blocks, each comprising 108 trials: two blocks involved the Search task, and another two blocks involved the NoSearch task. Search and NoSearch block pairs were presented consecutively, however their order was counterbalanced across participants. Within each pair of Search and NoSearch blocks, one block required the participant to monitor the upper row of houses, and the other required them to monitor the lower row of houses.

Each block comprised 27 trials from each condition (i.e., RJA, RJAc, IJA, IJAc). Social (RJA, IJA) and control (RJAc, IJAc) trials were presented in clusters of six trials throughout each block. Each cluster began with a cue lasting 1,000 ms that was presented over the avatar stimulus and read “Together” for a social cluster and “Alone” for a control cluster. Trial order randomisation was constrained to ensure that the location of the burglar, the location of blue doors, and the number of gaze shifts made by the avatar were matched within each block and condition.

After playing the interactive game, and consistent with our previous studies employing this paradigm, a post-experimental interview was conducted in which participants were asked to rate their subjective experience during the task (cf. Caruana, Brock & Woolgar, 2015,Caruana et al., in press). Full details on the assessment of subjective experiences and relevant findings are provided in Supplemental Information 2.

Eye tracking

Eye-movements from the right eye only were recorded with a sampling rate of 500 Hz using a desktop-mounted EyeLink 1000 Remote Eye-Tracking System (SR Research Ltd., Ontario, Canada). Head movements were stabilised using a chinrest. We conducted an eye tracking calibration using a 9-point sequence at the beginning of each block. Seven gaze areas of interest (GAOIs) over the houses and avatar stimulus were used by our gaze-contingent algorithm (see Caruana, Brock & Woolgar, 2015 for details). A recalibration was conducted if the participant made consecutive fixations on the borders or outside the GAOIs. Trials requiring a recalibration were excluded from all analyses.

Scores

Accuracy

We calculated the proportion of trials where the participant succeeded in catching the burglar in each condition (i.e., RJA and RJAc) for each task separately (i.e., Search and NoSearch). We excluded from the accuracy analysis any trials that required a recalibration or (in the Search task) were failed due to an error during the search phase.

Saccadic reaction times

For correct trials, we measured the latency (in ms) between the presentation of the gaze cue (for RJA trials) or the arrow cue (for RJAc trails), and the onset of the participant’s responding saccade towards the burglar location (see Fig. 2, Reaction time period).

Statistical Analyses

Saccadic reaction times were analysed via repeated measures Analysis of Variance (ANOVA) using the ezANOVA (ez) package in R (Lawrence, 2013), reporting the generalised eta squared ηG2 measure of effect size. Significant task*condition interactions effects were followed-up with Welch’s two sample unequal variances t-tests (Welch, 1947). As in our previous studies, we report analyses of the mean reaction time, having excluded trials with reaction times less than 150 ms as these are typically considered to be anticipatory responses. Trials timed out after 3,000 ms providing a natural upper limit to reaction times. Full syntax and output for this analysis can be found in the Rmarkdown document (Supplemental Information 1). The RMarkdown also provides complementary ANOVAs of the mean and median of the untrimmed data, as well as a mixed random effects analysis using the lme4 R package (Bates, 2005). The results of all analyses are consistent in terms of the predicted interaction between task and condition. A significance criterion of p < 0.05 was used for all analyses.

Results

As depicted in Fig. 3, participants performed at close-to-ceiling levels in terms of the trials successfully completed across all conditions. Of the small number of errors made, the majority were Location Errors in the RJA and RJAc conditions, whereby participants looked first to an incorrect location (house) rather than following the avatar’s gaze to the burglar location. Given the low number of errors, we do not report statistical analyses of accuracy or errors.

Figure 3 Plots of accuracy data by condition.

Box plots displaying the proportion of correct trials, proportion of time-out errors, and proportion of location errors. Data points represent individual participant means.

Figure 4 shows mean saccadic reaction times for correct trials in the RJA and RJAc conditions. Participants were significantly slower on the Search task than the NoSearch task (main effect of task (F(1, 15) = 11.07, p = .005, ηG2=0.13). They were also significantly slower to respond on RJA trials than RJAc trials overall (main effect of condition, F(1, 15) = 98.75, p < .001, ηG2=0.57). Importantly, there was also a significant task*condition interaction (F(1, 15) = 43.86, p < .001, ηG2=0.18), indicating a larger effect of task in the RJA condition. This interaction was also present in all re-analyses of the data (see Supplemental Information 1).

Figure 4 Plots of saccadic reaction time data.

Box plots displaying saccadic reaction times in RJA and RJAc conditions, separated by task (i.e., Search, NoSearch). Data points represent individual participant means.

Follow-up paired t-tests revealed that responses to social gaze were significantly slower in the Search task than the NoSearch task (t(15) = 4.82, p < .001), whereas response times to arrow cues did not significantly differ between the two versions of the control task (t(15) =  − 0.85, p = .411). Consistent with our previous studies, there was an effect of condition (i.e., slower responses to gaze cues than arrow cues) for the Search (t(15) = 9.31, p < .001) task. However, there was also a significant (albeit smaller) effect of condition for the NoSearch task (t(15) = 8.51, p < .001).

Discussion

One of the main challenges facing social neuroscience—and the investigation of joint attention in particular—is the need to achieve ecological validity whilst maintaining experimental control. During genuine joint attention experiences, our social cognitive faculties are engaged whilst we are immersed in complex interactions consisting of multiple social cues with the potential for communication. A critical but neglected aspect of joint attention is the requirement to identify those cues that are intended to be communicative. In the specific case of eye gaze cues, the responder must differentiate gaze shifts that signal an intentional joint attention bid from other, non-communicative gaze shifts. The results of the current study indicate that this intention monitoring process has a measurable effect on responsive joint attention behaviour.

The Search version of our Catch-the Burglar task was identical to that used in our previous studies. In the social (RJA) condition, participants found all of their houses to be empty, waited for their partner, Alan, to complete his search, make eye contact, and then guide them to the burglar’s location. We replicated our previous finding (Caruana, Brock & Woolgar, 2015; Caruana et al., in press) that participants were slower to respond in this condition than in the matched control (RJAc) condition in which the avatar’s eye gaze cue was replaced by an arrow.

The critical innovation of the current study was the addition of a NoSearch condition in which the same gaze and arrow cues were used but the joint attention episode was not preceded by a search phase, thereby removing the intention monitoring component of the RJA condition. Participants were still slower to respond to eye gaze than to arrow cues, suggesting that the previously identified difference between RJA and RJAc in the Search condition is not entirely attributable to intention monitoring. As discussed earlier, it is possible that differences in the perceptual salience of the arrow cue might help contribute this effect. Alternatively, participants may be affected by the presence of a social partner. The current study was designed to control for such factors and it is not possible to determine which if either of these explanations is correct.

The important finding was the task by condition interaction. This arose because the magnitude of the condition effect was significantly reduced in the NoSearch version of the task. This cannot be explained in terms of perceptual salience or social context, because these were identical across Search and NoSearch tasks.

The interaction can also be viewed by contrasting the effect of task (Search vs. NoSearch) for the two different conditions (RJA or RJAc). In the RJA condition, participants were significantly faster to respond to the eye gaze cue when the search phase was removed. The search phase required the participant and their virtual partner to make multiple non-communicative eye-movements prior to the joint attention episode. Participants therefore had to differentiate between eye-movements made by the avatar that signalled a communicative joint attention bid and those that were merely a continuation of their search. In the NoSearch task, every eye-movement made by the avatar was communicative, thereby removing the requirement to monitor his communicative intent, enabling faster response times.

Importantly, there was no effect of task (Search vs. NoSearch) for the RJAc condition. This allows us to discount a number of alternative explanations for the task effect in the RJA condition. For example, it could be argued that the slower responses in the Search task reflected differences in the timing of the stimulus presentation (e.g., the delay between participants fixating on the avatar and the avatar making his guiding saccade). However, the timing of the stimuli were programmed to be identical in the corresponding RJA and RJAc conditions of the Search and NoSearch tasks, so any effect of stimulus timing should have been evident in both conditions. Another plausible explanation is that participants were slower in the Search task because this required them to switch from searching for the burglar to responding to the avatar on each trial. But again this applied equally to the RJA and RJAc conditions, so it cannot explain the task by condition interaction.

In short, the observed interaction between task and condition is entirely consistent with our intention monitoring account and cannot be explained in terms of the perceptual salience of different cues, the task’s social context, the timing of the stimulus presentation, or the requirement to switch between searching and responding.

The current data provide insights into our other recent findings in studies using the Search version of our interactive task. In one study, we used fMRI to investigate the neural correlates of RJA (Caruana, Brock & Woolgar, 2015). By contrasting activation in the RJA (eye gaze) and RJAc (arrow) conditions, we identified a broad frontotemporoparietal network including the right temporo-parietal junction and right inferior frontal lobe. These brain regions are strongly associated with aspects of social cognition including mentalising (e.g., Saxe & Kanwisher, 2003) and predicting anothers’ actions (Danckert et al., 2002; Hamilton & De Grafton, 2008) but have not been previously linked to RJA (cf. Redcay et al., 2012; Schilbach et al., 2010). The tasks used in previous studies of RJA were similar to the current NoSearch task. As such, they would not have captured the intention monitoring processes involved in RJA, perhaps explaining the discrepenacy with our fMRI study. Future neuroimaging studies of joint attention could employ the current study’s design, and compare activation observed during the Search and NoSearch task. If our interpretation is correct then removing the search component should reduce the involvement of temporoparietal and inferior frontal regions in the RJA condition.

In another study (Caruana et al., in press), we investigated joint attention in adults with autism. Observational studies of real-life interactions provide overwhelming evidence that joint attention impairments are a core feature of autism (Charman et al., 1997; Dawson et al., 2004; Loveland & Landry, 1986; Mundy, Sigman & Kasari, 1990; Osterling, Dawson & Munson, 2002; Wong & Kasari, 2012). However, previous computer-based experimental studies of joint attention have largely failed to find consistent evidence of gaze following difficulties (Leekam, 2015; Nation & Penny, 2008). One possible explanation for this is that autistic individuals have an underlying difficulty in understanding the social significance or communicative intentions conveyed by eye contact (cf. Böckler et al., 2014; Senju & Johnson, 2009) that is not captured by the tasks used in previous studies of autism. In contrast to these studies, we did find evidence of impairment: autistic adults made more errors and were slower to respond than control participants in the RJA condition despite showing no impairment on the control condition. Future studies involving individuals with autism and the Search and NoSearch versions of our task would clarify this issue further.

Another avenue for further research using this task would be to investigate the development of joint attention in young children. Infants begin responding to and initiating joint attention bids within the first year of life (Mundy et al., 2007), but virtual reality tasks provide the sensitivity to investigate the developmental changes in the speed and efficiency of joint attention engagement in later development. Finally, it would be of interest to investigate sex differences in performance. Studies of infants (Saxon & Reilly, 1999) and school-aged children (Gavrilov et al., 2012) have found that females exhibit increased joint attention behaviours compared to their male peers, although it is unclear to what extent these differences reflect underlying differences in competence as opposed to motivation. A limitation of the current study is that only three participants were female. However, future studies with larger samples would allow systematic investigation of sex differences in joint attention performance at multiple points across development.

Summary

In everyday joint attention episodes, a critical aspect of responding to joint attention bids is the ability to discern which social cues have communicative intent and which do not. The results of the current study indicate that this intention monitoring component has a measureable effect on responding behaviour. Moreover, this component can be isolated by contrasting joint attention episodes occurring in the context of a realistically complex social interaction versus a simplified context in which each cue is unambiguously communicative. The clear differences in performance on the Search and NoSearch versions of our task highlight the importance of striving for ecological validity in studies of social cognition (cf. Schilbach et al., 2013). The results also demonstrate the potential of our task for investigating the different components of joint attention in typically developing children and in clinical populations associated with atypical social cognition.

Supplemental Information

Supplemental Information 1 RMarkdown of Statistical Analyses and Results

Click here for additional data file.

Supplemental Information 2 Subjective Task Ratings

Click here for additional data file.

Additional Information and Declarations

Competing Interests

Author Contributions

Human Ethics

Data Availability

Genevieve McArthur and Jon Brock are Academic Editors for PeerJ

Nathan Caruana conceived and designed the experiments, performed the experiments, analyzed the data, contributed reagents/materials/analysis tools, wrote the paper, prepared figures and/or tables, reviewed drafts of the paper.

Genevieve McArthur and Alexandra Woolgar conceived and designed the experiments, wrote the paper, reviewed drafts of the paper.

Jon Brock conceived and designed the experiments, analyzed the data, contributed reagents/materials/analysis tools, wrote the paper, prepared figures and/or tables, reviewed drafts of the paper.

The following information was supplied relating to ethical approvals (i.e., approving body and any reference numbers):

Macquarie University Human Research Ethics Committee, approval reference number: 5201200021.

The following information was supplied regarding data availability:

Open Science Framework: osf.io/6e597.

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
