# Peer review of "Detecting communicative intent in a computerised test of joint attention"

_PeerJ, doi:10.7717/peerj.2899_

## Round 0.1 · original submission · Minor Revisions

I have now received two reviews from experts in your area. I thank the reviewers for their work. As you will see, both reviewers are positively inclined toward your work (minor revision) and I share their positive evaluation.

In the rest of my action letter I will focus on the aspects that I consider crucial for a successful revision. For the remaining parts, I invite you to refer to the reviews, that are thoughtful and very clear.

Method. The control condition could be explained more clearly.
Error rate. You could explain in light of similar studies the reasons why the error rate is so high.

Social attention. You could discuss whether further mechanisms regulating attention in social context can at least partially explain your data (see comments of Reviewer 2).

Limitations. Possible limitations of the study (e.g. the sample selection) could be further discussed (see comments of Reviewer 1).

·

Basic reporting

No comments

Experimental design

The research question is clear and it fills an identified knowledge gap. The method implemented in the experiment is appropriate, and in general the authors described the experimental procedure with sufficient information to replicate it. However,while for the social condition (i.e., avatar) the instructions are clear, they are not for the arrow one. Specifically, for the arrow conditions, it is not clear if participants were informed or not that the searching phase was pc-controlled.

Validity of the findings

The findings reported in the paper are interesting and add nicely to the growing literature on the relation between intention monitoring processes and joint attention behaviour. Results are well discussed and they answer the main research question. I only have some minor comments:
- When describing the control condition (i.e. arrow cues) page 13 line 247, the authors write:” participants were told that they would play the game without Alan, whose eyes remained closed during the trial”

It is not clear whether participants were explicitly informed that the searching phase was controlled by an algorithm, please make clear this point by specifying how the searching phase was described in the control condition.

- In the results section, the authors reported the results for responding condition only. For sake of clarity and to increase the reproducibility of the paradigm, please report in the results session also results from the Initial condition.

- The sample is not homogeneous for gender. Indeed, out of 16 subjects only 3 females were tested. Given the massive literature reporting gender differences in joint attention and social intention processing, I think that the authors should point out this limit of the study in the discussion section.

·

Basic reporting

No Comments.

Experimental design

No Comments.

Validity of the findings

No Comments.

Additional comments

This manuscript examined the influence of intention monitoring on the reaction time (RT) of gaze-following response during joint attention. Participants engage with a virtual avatar to collaboratively locate a target in a visual search array. Two task conditions were created: the Search condition requires participants to differentiate communicative gaze from searching gaze while the NoSearch condition involves only communicative gaze. It is found that RT was larger in the Search condition than the NoSearch condition, demonstrating the influence of intention monitoring. The same RT effect was not observed in a non-social control experiment where avatar’s gazes are replaced by arrows.

Overall, this is a well-written manuscript. The motivation of this study is clearly articulated; the experimental design is valid and the data quality is high. The findings are important and of broad interest to the field. However, I would like to raise a few concerns that should be addressed by the authors.

Major concern:

The authors appear to have ignored a very important aspect of their data in the discussion: NoSearch RTs are significantly larger in the RJA than RJAc trials (>100ms). The aim of the study was to identify the mechanisms underlying an approximately 200ms RT difference between avatar condition and arrow condition (Caruana et al 2015). If larger RT was purely caused by the process of distinguishing communicative gaze from non-communicative gaze, then NoSearch trials in the avatar condition should be as fast as the arrow condition, because there is no cue ambiguity in both cases. This is not the case in their data. In fact, this manipulation explains about half of the RT differences. Clearly, something else is going on which I think is worthwhile to mention and should be discussed adequately. Identifying communicative gaze represents only one of the many aspects of intention monitoring during social interaction. For example, it is shown that observing an action (gaze shift) triggers complementary (physical and mental) aspects of action processing (De Lange et al 2008 Curr Biol; Kuang 2016, Front Hum Neurosci). There is also other information (emotion, dominance, etc) available during face perception which can attract social attention and influence RT. A recent paper has reviewed various aspects of attention in social contexts (Kuang 2016 Front Psychol), and should be discussed here to help explain the remaining RT effects (100ms) between RJA and RJAc in the NoSearch condition.

Minor questions:

1. Introduction (line 134): Please clarify “the arrow cue was more salient than the corresponding gaze cue”. Be specific about which aspects of the arrows are more salient.

2. Methods (line 307-309): The authors excluded trials with anticipatory response (less than 150ms). This is fine, but how was the cutting-threshold decided (150ms)? Did the authors try other critiera, e.g., mean±3std, to exclude outlier trials? This might affect mean RT considerably, given that the time out was set exceptionally high (3000ms). Ideally, the authors should have checked the distribution of RTs to have a better sense of the data. This is critical because RT is the only core result that supports their conclusions.

3. Results (page 17, bottom paragraph): Please indicate the nature of error trials. Was it more often by incorrect judgments, or by time-out (longer than 3000ms)? I feel that the error rate was relatively too high, given the simplicity of the task and the long response time.

---

## Round 0.2 · Minor Revisions

While Reviewer 1 is satisfied with your revision, Reviewer 2 suggests some minor modifications. Please proceed in answering to his/her comments. I am looking forward to your revision.

·

Basic reporting

The authors made all the requested changes and they clarified all the methodological aspects.

Experimental design

The method implemented in the experiment is appropriate, and the authors clarified all the methodological aspects.

Validity of the findings

Results section is clear and it now reports all the information to reproduce the experiment and compare the results with those of similar studies.

In the section: SUPPLEMETARY MATRIAL 2 pag. 2 line 3, the authors write "the were no significant differences between the serch and not search tasks on ratings of task difficulty, naturalness, intuitiveness, pleasantness (all ps > .054)". However, at page 3 the p-values reported in Table 1 are all greater than .066. Please check the values reported in Table 1 and report the correct lower p-value in the text.

·

Basic reporting

none.

Experimental design

none.

Validity of the findings

none.

Additional comments

I applaud the authors for having made a thorough revision effort addressing each of my previous concerns, which has improved the strength of this manuscript considerably. Nevertheless, I would like to add one more piece of comment regarding the social attention argument. It appears to me that the authors might have just misunderstood my point slightly.

From the revision and the added citation (Wykowska et al 2014), it seems that the authors considered social attention narrowly in the strict sense that it depends on whether participants believe they are interacting with real humans or computers. But I was referring to social attention in a broader sense that it operates at the lower perceptual level and that it is independent of the presence of social partnership. To give you an example, it has been repeatedly shown that computerized static face picture (with gaze direction manipulated) triggers reflective visuospatial orienting (Driver et al., 1999; Friesen & Kingstone, 2003; Langton & Bruce, 1999; Ricciardelli, Bricolo, Aglioti, & Chelazzi, 2002), and these reflexive attentional effects are reliably present irrespective of whether the faces are simple schematic faces or more realistic face pictures (Tipples, 2005). For a extensive review on this topic please refer to (Frischen, Bayliss, & Tipper, 2007) paper. In all these studies participants had no sense of interacting with real human agents but there are robust attentional effects attracted to the social signal. The social signals are present not only in low-level face/gaze perception but also in low-level action recognition. For instance, even point-light displays mimicking the motion of a living organism can also capture reflexive attentional orienting (Johansso.G, 1973; Shi, Weng, He, & Jiang, 2010; Troje, 2002; Blake & Shiffrar, 2007). Similarly, mirroring and mentalizing processes can be present not only during real social interactions but also during watching action videos/clips. To sum up, my point is, these basic perception and action related attentional processes likely exist in the avatar stimuli used in the current study, which might account for the ~100 ms RT differences between RJA and RJAc trials. I therefore suggested the three most recent publications (including two of my own) on these issues that help to cover these ideas.

Reference List

Blake, R., & Shiffrar, M. (2007). Perception of human motion. Annu Rev Psychol, 58, 47-73.
Driver, J., Davis, G., Ricciardelli, P., Kidd, P., Maxwell, E., & Baron-Cohen, S. (1999). Gaze perception triggers reflexive visuospatial orienting. Visual Cognition, 6(5), 509-540.
Friesen, C. K., & Kingstone, A. (2003). Abrupt onsets and gaze direction cues trigger independent reflexive attentional effects. Cognition, 87(1), B1-10.
Frischen, A., Bayliss, A. P., & Tipper, S. P. (2007). Gaze cueing of attention: visual attention, social cognition, and individual differences. Psychol Bull, 133(4), 694-724.
Johansso.G. (1973). Visual-Perception of Biological Motion and a Model for Its Analysis. Perception & Psychophysics, 14(2), 201-211.
Langton, S. R. H., & Bruce, V. (1999). Reflexive visual orienting in response to the social attention of others. Visual Cognition, 6(5), 541-567.
Ricciardelli, P., Bricolo, E., Aglioti, S. M., & Chelazzi, L. (2002). My eyes want to look where your eyes are looking: exploring the tendency to imitate another individual's gaze. Neuroreport, 13(17), 2259-2264.
Shi, J., Weng, X., He, S., & Jiang, Y. (2010). Biological motion cues trigger reflexive attentional orienting. Cognition, 117(3), 348-354.
Tipples, J. (2005). Orienting to eye gaze and face processing. J Exp Psychol Hum Percept Perform, 31(5), 843-856.
Troje, N. F. (2002). Decomposing biological motion: a framework for analysis and synthesis of human gait patterns. J Vis, 2(5), 371-387.

---

## Round 0.3 · accepted · Accept

I think your article represents and interesting and novel contribution to the field.